# Cuffless Blood Pressure Estimation Using Pressure Pulse Wave Signals

**DOI:** 10.3390/s18124227

**Published:** 2018-12-02

**Authors:** Zeng-Ding Liu, Ji-Kui Liu, Bo Wen, Qing-Yun He, Ye Li, Fen Miao

**Affiliations:** 1Shenzhen Institutes of Advanced Technology, Chinese Academy of Sciences, Shenzhen 518055, China; zd.liu@siat.ac.cn (Z.-D.L.); qy.he@siat.ac.cn (Q.-Y.H.); ye.li@siat.ac.cn (Y.L.); 2Shenzhen College of Advanced Technology, University of Chinese Academy of Sciences, Shenzhen 518055, China; jk.liu@siat.ac.cn (J.-K.L.); bo.wen@siat.ac.cn (B.W.); 3Key Laboratory for Health Informatics of the Chinese Academy of Sciences (HICAS), Shenzhen Institutes of Advanced Technology, Shenzhen 518055, China

**Keywords:** cuffless blood pressure, piezoelectric sensor, pressure pulse waveform, pulse transit time, multiparameter fusion

## Abstract

Pulse transit time (PTT) has received considerable attention for noninvasive cuffless blood pressure measurement. However, this approach is inconvenient to deploy in wearable devices because two sensors are required for collecting two-channel physiological signals, such as electrocardiogram and pulse wave signals. In this study, we investigated the pressure pulse wave (PPW) signals collected from one piezoelectric-induced sensor located at a single site for cuffless blood pressure estimation. Twenty-one features were extracted from PPW that collected from the radial artery, and then a linear regression method was used to develop blood pressure estimation models by using the extracted PPW features. Sixty-five middle-aged and elderly participants were recruited to evaluate the performance of the constructed blood pressure estimation models, with oscillometric technique-based blood pressure as a reference. The experimental results indicated that the mean ± standard deviation errors for the estimated systolic blood pressure and diastolic blood pressure were 0.70 ± 7.78 mmHg and 0.83 ± 5.45 mmHg, which achieved a decrease of 1.33 ± 0.37 mmHg in systolic blood pressure and 1.14 ± 0.20 mmHg in diastolic blood pressure, compared with the conventional PTT-based method. The proposed model also demonstrated a high level of robustness in a maximum 60-day follow-up study. These results indicated that PPW obtained from the piezoelectric sensor has great feasibility for cuffless blood pressure estimation, and could serve as a promising method in home healthcare settings.

## 1. Introduction

Blood pressure (BP) is a vital physiological parameter for clinical diagnosis and treatment of hypertension, with values varying between systolic blood pressure (SBP) and diastolic blood pressure (DBP). Hypertension is one of the key risk factors for cardiovascular disease (CVD), as it is reported to be the primary cause of death and disability worldwide [1]. The World Health Organization reported that the hypertension prevalence is 24% and 20.5% in men and women, respectively [2]. However, because most hypertensive patients are not aware of their disease, and it damages their internal organs without them knowing, hypertension is called the silent killer [3]. In addition, BP variability has been reported to be a valuable prognostic indicator for hypertension and CVD [4]. Therefore, the use of a non-obstructive device for monitoring continuous BP is crucial for early prevention, diagnosis, and treatment of hypertension.

The most precise method to continuously measure BP is by using arterial cannula, in which a catheter is inserted into the blood vessel [5], but this invasive method can cause complications. A traditional 24-h ambulatory BP device adopts the auscultatory or oscillometric method to monitor BP at regular intervals. However, this method is discontinuous and causes discomfort, due to repeated cuff inflations. Several technologies, such as tonometry and volume clamps, have also been used for continuous BP measurement; however, they are expensive and uncomfortable for the patient in long-term monitoring during daily activities [6]. Pulse transit time (PTT) is a potential indicator to estimate BP [7], and it refers to the time taken by the arterial pulse traveling from the heart to the peripheral arterial site [8]. The BP estimation method based on PTT has received considerable attention because it can offer a cuffless and comfortable long-time BP measurement, with no risk to the patient [8,9,10]. However, this method required at least two simultaneous signals, such as electrocardiogram (ECG) and pulse wave, which is hard to deploy in wearable devices for cuffless and continuous BP estimation [11].

Cuffless BP measurement based on a one-channel pulse wave is an effective approach to reduce the complexity and inconvenience of PTT-based methods. Generally, there are optical and pressure sensing technique in pulse wave measurement. For optical technique, a photoelectric sensor is used to measure a photoplethysmogram (PPG) by calculating the light absorption changes induced by blood volume changes in arterioles [12]. Pressure sensing technique measures a pressure pulse wave (PPW) that reflects the pressure changes within the blood vessel in the artery by using a pressure sensor (e.g., piezoelectric sensor) [13]. Several studies have discussed cuffless BP estimation using PPG signals based on optical sensor [12,14,15]. For example, it has been reported that multiparameters extracted from single PPG signals have the potential to estimate BP [14]. Lin et al. [9] verified that a combination of features calculated using PPG can improve the PTT-based method for cuffless blood pressure estimation; however, the accuracy of BP estimation models based on only PPG features was inferior to that of the PTT-based model in this method. Moreover, most of PTT- and PPG-based studies were verified on young healthy subjects, and thus are limited while applied to a diversified population. In another aspect, PPG measured from an arteriole (e.g., finger and ear) reflects the blood volume changes in the microvascular bed of tissue [16]. Therefore, the PPG amplitude is usually weak and can easily be interfered with by motion artifact, respiration effect, and low perfusion [17]. The process of calculating features from PPG waveforms is complicated and error-prone, due to the influence of peripheral microvascular tissue resistance. As a result, the accuracy of BP estimation based on PPG may be unstable and unreliable. Compared with PPG signals that are attenuated due to peripheral vasoconstriction [18], PPW signals obtained from piezoelectric sensors have less interference and more abundant characteristic information, and have been well-studied for disease diagnosis and arterial stiffness evaluation [19,20]. However, few studies exist regarding the utility of PPW for cuffless BP estimation. PPW can be collected in a wearable manner with the development of electric fabric and flexible pressure sensors [21,22], which makes the utilization of PPW for cuffless and wearable BP estimation more feasible.

In this paper, we investigated the modeling of BP from PPW for cuffless BP estimation based on the piezoelectric technique. This study makes several contributions to the cuffless BP measurement fields. First, this study presents a comprehensive evaluation on the performance of PPW signals measured by piezoelectric sensor in BP estimation for the first time to our best knowledge, various BP indicators are extracted from PPW, and their performances in estimating BP are investigated and compared with that of existing PTT features. Second, a number of middle-aged and elderly participants (including hypertensive subjects and hypotensive subjects) are involved in our study and followed up on for a maximum of 60-day period to verify the performance of the model, thus indicating the high reliability of the proposed approach.

## 2. Background

### 2.1. Principle of PPW Measurements Based on Piezoelectric Sensors

Periodic contraction and relaxation of the heart creates rhythmic intermittent blood ejection, which leads to pulsation changes in the pressure of the artery, and then causes action–reaction processes such as vibrations of the skin surface [23]. The pulsation phenomenon can be sensed while pressing fingers on the superficial artery. In Traditional Chinese Medicine, the practitioners take the pulse by feeling the pulsations at radial artery of the wrist with certain amount of pressure to diagnose health condition of a subject [24].

With the advancement of modern sensor advances in sensor technology, the pulsation changes can be recorded by a pressure sensor such as capacitive, piezoresistive, and piezoelectric device. The piezoelectric sensor has a diaphragm structure, which is an ideal design for measuring fluctuating input pressure signals [25]. It has been widely used in PPW detection [26,27,28]. Figure 1 shows the principle of a piezoelectric sensor measuring PPW. Piezoelectric sensors use piezoelectric-sensitive materials such as polyvinylidene fluoride (PVDF) film, which contacts the human body skin directly to measure artery pulses by generating charge changes by itself in responding to mechanical pressure applied on the material [25] (Figure 1a), and converting the sensed results (charge changes) into electric signals by a charge amplifier [26] (Figure 1b). Finally, the output electrical signals of piezoelectric sensor are PPW (Figure 1c). As an obvious superficial artery, the radial artery of wrist is the most common site to measure PPW. It is worth mentioning that arterial tonometry can also be used to measure the BP waveform within the blood vessel, which need a controlled force to maintain the superficial artery in an applanated state over time [6,29]. However, applanation have proven to be difficult, so that the measured waveform should be frequently calibrated in practice [30]. Different from arterial tonometry, piezoelectric sensors do not need to flatten the artery, therefore the contact pressure is not quite restricted for PPW collection.

### 2.2. Relationship between PPW and BP

A typical PPW waveform is a composite of the forward wave created when the left ventricle contracts, and the reflected wave is produced when the forward wave is reflected from the peripheral vessels [31]. It comprises an ascending branch and a descending branch, as shown in Figure 1c, which is similar to the arterial blood pressure (ABP) wave contour. However, PPW cannot be directly transformed into ABP because of the nonlinearity of the blood flow, and the elasticity of the arterial wall.

Pulse waveform analysis based on PPW is an effective diagnostic tool in medical applications. Researchers have investigated the link between the PPW character and the cardiovascular system functional status [32,33,34]. With changes in physiological factors such as peripheral resistance, vascular wall elasticity, and blood viscosity, the PPW morphological characters have been reported to change regularly. When the peripheral resistance decreases and cardiac output increases, the amplitude and slope of the ascending branch increase. Conversely, when the peripheral resistance increases and the cardiac output decreases, the amplitude and slope of the ascending branch decreases. A similar situation occurs on the descending branch. In addition, the reflected wave arrives back at the central arteries earlier, and the amplitude of the reflected wave increases as the arterial stiffness increases, causing augmentation of the systolic pressure and a consequent decrease in diastolic pressure [35]. These studies verified that PPW may be used for noninvasive BP analysis.

## 3. Methodology

### 3.1. Experimental Protocol

A total of 65 participants (29 men and 36 women) were recruited in this study, with a mean age of 49.74 ± 8.12 years. Among them, eight subjects had hypertension (mean SBP greater than 140 mmHg or mean DBP greater than 90 mmHg over three-time BP collection via oscillometric technique) and 10 subjects had hypotensive (mean SBP less than 90 mmHg or mean DBP less than 60 mmHg over three-time BP collection via the oscillometric technique). None had taken antihypertensive drugs recently before the study was conducted. In the experiments, each participant was asked to lie down quietly to avoid motion artifacts. Figure 2 presents the experimental scenario. The ECG signals were acquired in this experiment for calculating the PTT. The reference BP (SBP and DBP) was measured using a cuff-based blood pressure device (HEM-7200, OMRON, OMRON Industrial Automation Japan, Kyoto, Japan) that was worn on the right upper arm, and 1-min ECG and PPW were continuously and simultaneously acquired using a custom-made multiparameter monitor system immediately after the BP measurement. The sampling frequency was set to 2500 Hz. ECG signals were recorded with electrodes placed on the left and right arms. A PPW evaluation module (HK-2000B, Hefei Huake Electronic Technology Research Institute, Hefei, China) based on piezoelectric sensitive elements was pressed on the radial artery of the left wrist to obtain PPW signals. The HK-2000B is a medical pulse sensor that integrates pressure-sensitive devices (PVDF piezoelectric membrane), sensitivity temperature compensation components, a temperature sensing element, and signal adjustment circuits, and it has been recognized by some research institutions [36]. The major technique parameter of HK-2000B pulse sensor was summarized in Table 1. All participants were instructed to remain still during signal acquirement to suppress interference. Informed consent was obtained from all participants prior to the experiment, in accordance with the guidelines of the Institutional Ethics Board of Shenzhen Institutes of Advanced Technology, Chinese Academy of Sciences.

As shown in Figure 3, two sets were recorded over five days for each participant: a training set and a testing set. On the first day (T day), no less than 16 datasets evenly distributed between 9:00 a.m. and 5:00 p.m. were collected for the training set, and 16 datasets (with four datasets for each day at a random time) were collected during the next four days for the testing set. The first day and the next four days were not consecutive days, with different intervals between them: (T + 1), (T + 3), (T + 6), and (T + 8) present at 1, 3, 6, and 8 days after the training dataset was collected, respectively. For each participant, the training set was used to develop an individual BP model, whereas the independent testing set was used for examining the performance of the developed model.

Finally, 1216 datasets obtained in the training set, with ranges of 121.53 ± 15.97 mmHg in SBP and 72.65 ± 10.63 in DBP, and 1040 datasets obtained in the testing set, with ranges of 119.31 ± 15.55 mmHg in SBP and 70.76 ± 10.13 mmHg in DBP. Figure 4a presents the statistical information regarding the distribution of all SBP and DBP. The BP dynamic range for each participant was calculated here as the difference between the maximum BP and minimum BP in the whole measurement process. Figure 4b demonstrates the distribution of individual BP dynamic range, suggesting BP is significantly changed during the experiment process with a mean dynamic range of 24.97 mmHg for SBP and 17.65 mmHg for DBP.

### 3.2. Signal Processing

To avoid the unpredictable effects of noise and artifacts from raw signals, the signals were preprocessed to filter and denoise them. ECG signals and PPW signals were first preprocessed using a low-pass filter with a cutoff frequency of 50 Hz to remove high-frequency interference, and then the baseline drift of PPW was removed based on wavelet transformation [37]. Subsequently, numerous features were extracted from the denoised PPW and ECG signals.

### 3.3. Feature Extraction

Based on the pulse wave analysis, 22 (numbered from 1 to 22) informative features were calculated from each beat of PPW and ECG. The first derivative of PPW (1st dPPW), the second derivative of PPW (2nd dPPW), and the ECG signals that potentially influenced BP were analyzed in the present study by using one PTT feature for comparison and 21 other PPW features, as shown in Table 2, and Figure 5 and Figure 6.

Figure 5 illustrates the definitions of the extracted features. In Figure 5 and Table 1, letters “A”, “C”, “D”, “E” represent the foot, peak, foot of the dicrotic wave, and peak of the dicrotic wave in PPW, respectively; letters “F”, “G”, and “H” represent the start, peak, and foot in the 1st dPPW; letters “N”, “M”, and “L” represent the start, peak, and foot in 2nd dPPW. Point B in PPW is the maximum point of the 1st dPPW. PTT was calculated as the time intervals from the ECG R-peak to the peak of the 1st dPPW in the same cardiac cycle [8]. Additionally, the 3 K-values are shown in Figure 6, which were calculated using the Formulas (1)–(3). For dicrotic wave detection, after obtaining the point “A” (maximum point) in PPW, the zero-crossing points behind “A” within the specific range were examined to find the points “D” and “E” in the dicrotic wave during a normal state. If this was not possible, the maximum point of the 1st dPPW behind the “A” point within the specific range was assigned to a feature point, and the points “D” and “E” coincide in this case.
(1)$$K=\frac{P_{m}-P_{d}}{P_{s}-P_{d}}$$
(2)$$K1=\frac{P_{m1}-P_{d}}{P_{s}-P_{d}}$$
(3)$$K2=\frac{P_{m2}-P_{d}}{P_{s}-P_{d}}$$where $P_{m}=\frac{1}{T}\int_{0}^{T}P(t)$, $P_{m1}=\frac{1}{t1}\int_{0}^{t1}P(t)$, $P_{m2}=\frac{1}{t2}\int_{t1}^{t2}P(t)$.

### 3.4. Data Analysis

#### 3.4.1. BP Estimation Models

For each feature, the average beat-to-beat feature during the 1-min recording was calculated for each dataset, and then corresponded to the reference BP measurement in that dataset. A BP estimation model was firstly constructed through univariate linear regression by using the least squares method for each feature. A total of 21 PPW-based BP estimation models were constructed based on each of the above 21 features (numbered 1–21, as listed in Table 2) as an independent variable, and then a multiparameter fusion (MPF) model, based on the sample-trimmed mean and the standard deviation of those 21 PPW-based models, was developed for BP estimation. The sample-trimmed mean is a robust estimate of the sample location, as it can address the problems of heavy-tailed distribution in the sample data [38], and the standard deviation is a measure that reflects the amount of variation or dispersion of a set of sample data values [39]. Let $F_{1},F_{2},\ldots ,F_{21}$ denote the BP estimation values of the 21 PPW-based models respectively, and let $F_{\left(1\right)},F_{\left(2\right)},\ldots ,F_{\left(21\right)}$ be the observations written in ascending order. The MPF model based on the sample-trimmed mean and standard deviation can be calculated as:(4)$$\mathrm{MPF}=\frac{F_{\left(g+1\right)}+\cdots +F_{\left(N-g\right)}}{N-2g}-STD$$where N is the length of sample (i.e., N=21 in this paper), g=γ⋅N (0≤γ<5) is the desired amount of trimming, and STD is the standard deviation of $F_{1},F_{2},\ldots ,F_{21}$.

Thus, total 22 PPW-based models (numbered 1–22, as listed in Table 3) were developed for BP estimation. The performance of the PPW-based models was compared with three previous reported PTT-based models: (1) a linear model developed based on PTT [40]; (2) one of the most widely used models based on 1/PTT [41]; (3) BP estimation model combined with ln(1/PTT) and 1/PTT^2^ [42,43].

#### 3.4.2. Performance Assessment of BP Estimation Models

We evaluated the performance of the established BP models by measuring the mean difference (MD) and standard difference (SD) of the estimation errors across all of the participants (totally 1040 datasets). The MD and SD were defined by Equations (5) and (6), respectively. The cumulative percentage (CP) of BP estimation errors by the proposed models falling within the values of ±5 mmHg, ±10 mmHg and ±15 mmHg was calculated according to the British Hypertension Society standard (BHS) [44], and then the BHS, as well as the US Association of Advancement of Medical Instrumentation standard (AAMI), was used to evaluate the performance of the proposed BP estimation models. A two-sample t-test was utilized to test the estimation errors of different model, with *p* < 0.05 being regarded as statistically significant. In addition, the Pearson’s correlation and Bland-Altman analysis between estimated BP from established model and reference BP was calculated as well.
(5)$$MD=\frac{\sum_{i=1}^{n}\left(y_{i}-x_{i}\right)}{n}$$
(6)$$SD=\sqrt{\frac{\sum_{i=1}^{n}{\left(y_{i}-x_{i}-MD\right)}^{2}}{n-1}}$$where $\left\{x_{1},x_{2},\cdots ,x_{n}\right\}$ are the reference BP values obtained from the cuff-based device and $\left\{y_{1},y_{2},\cdots ,y_{n}\right\}$ are the estimated BP values obtained from the constructed BP model.

## 4. Experimental Results

### 4.1. Performance of the BP Estimation Models

Table 3 summarizes the MD and SD errors for the PPW-based BP estimation models and the PTT-based model tested on 65 participants, including 1040 datasets, with reference cuff-based BP. The proposed MPF model (model 22, with 0.70 ± 7.78 mmHg and 0.83 ± 5.45 mmHg for SBP and DBP estimation errors, respectively) showed a better performance in BP estimation among all the PPW-based models, and met the AAMI standard for accuracy requirements of 5 ± 8 mmHg (MD ± SD error) under the present experimental scenario. Compared with the best existing PTT-based model (model 25, with 2.03 ± 8.15 mmHg, and 1.97 ± 5.75 mmHg for SBP and DBP estimation errors, respectively), the estimation errors reduced 1.33 ± 0.37 mmHg estimation errors in SBP and 1.14 ± 0.20 mmHg estimation errors in DBP. We evaluated the statistical significance of the differences between the estimation errors of the proposed model (model 22) and the PTT-based model (model 25), using a two-sample *t*-test. The results showed that the difference was statistically significant at the level of 0.0001, indicating that the features calculated from PPW have the potential to estimate BP and exhibit a more efficient performance than that of the existing PTT feature. In the rest of the paper, the MPF model (model 22) was used for further comparative analysis with the best PTT-based BP estimation model (model 25).

Table 4 presents the evaluation of the proposed MPF model and the existing PTT-based model based on the BHS standard. We can observe that the proposed MPF model complies with the BHS Grade B standard (CP at ± 5 mmHg > 50%, CP at ± 10 mmHg > 75%, CP at ± 15 mmHg > 90%) in the estimation of SBP and the Grade A standard (CP at ± 5 mmHg > 60%, CP at ± 10 mmHg > 85%, CP at ± 15 mmHg > 95%) in the estimation of DBP, which achieved a better accuracy than the PTT-based model.

Figure 7 provides a Pearson’s correlation coefficient (CC) and a Bland–Altman analysis between the overall reference BP and estimated BP values by the MPF model and the PTT-based model for all 65 participants. The CC between the estimated SBP and DBP of the MPF model and the cuff-based measurement was 0.87 (*p* < 0.0001) and 0.85 (*p* < 0.0001), respectively, which were higher than those of the PTT-based model (CC of 0.83 and 0.80 for SBP and DBP, respectively). The Bland–Altman plot indicates that the estimated BP by the proposed MPF method approximated the reference BP measured by cuff-based device.

Figure 8 and Figure 9 present a typical subject of the comparison between the reference BP and the estimated BP derived from different models. The SBP estimation errors were −0.33 ± 4.32 mmHg and 1.86 ± 4.86 mmHg for the MPF model and the PTT-based model, respectively. The corresponding DBP estimation errors were 0.35 ± 3.54 mmHg and 1.18 ± 3.71 mmHg. The estimated BP results from the MPF model showed a better performance for tracking the trend of BP than those of the PTT-based model.

### 4.2. Robustness Performance of the PPW-Based BP Models

To evaluate the robustness of the proposed BP estimation model, we compared the estimation accuracy at different time intervals for all 65 participants; that is, at (T + 1), (T + 3), (T + 6), and (T + 8) days after model construction. In addition, we investigated the estimation accuracy for 10 participants randomly selected from the 65 participants (T + 60) days after model construction to validate the long-term performance of the proposed model.

Figure 10 illustrates the trend in SBP and DBP estimation errors in the proposed MPF model compared with the PTT-based model. Note that the datasets at (T + 1), (T + 3), (T + 6), and (T + 8) day were obtained from 65 participants, and that of (T + 60) day were obtained from 10 participants. At (T + 1) day after construction, the overall errors of SBP and DBP estimations in the proposed MPF model were −0.23 ± 7.01 mmHg and 0.24 ± 4.60 mmHg respectively, which significantly increased to 1.14 ± 8.23 mmHg and 1.32 ± 5.89 mmHg at (T + 6) day, and to 1.25 ± 7.30 mmHg and 1.04 ± 4.91 mmHg at (T+8) days after model construction. The estimation errors were relatively stable for SBP and DBP from (T + 3) day (0.65 ± 8.60 mmHg errors and 0.53 ± 6.28 mmHg errors for SBP and DBP, respectively) to (T + 6) day, and then to (T+8) day. In addition, at (T + 60) day, the estimation errors of SBP and DBP in the proposed MPF model were 0.72 ± 5.33 mmHg and 1.62 ± 5.05 mmHg, respectively, within acceptable limits of error. These results indicated that the proposed MPF model had satisfactory robustness in BP measurement. The MPF model shows a similar robustness pattern as the PTT-based model but with consistent lower estimation errors on each follow-up day.

## 5. Discussion

### 5.1. PPW-Based Method for BP Estimation

PPW measured by the pressure sensing technique is another form of pulse wave, which not only provides information about intra-arterial pressure changes but also offers body cardiovascular system status and has the potential for the early diagnosis of vascular diseases [13,24]. PPW has more rich and obvious waveform characteristics than PPG, which makes it easier to accurately calculate parameters. Physiologically, the position of the dicrotic notch in a pulse wave holds valuable information [45]. It was reported that the features calculated from the dicrotic notch of PPG, such as the time span between the dicrotic and main peak of first derivative of PPG [11], and a large artery stiffness index [46], are effective indicators of arterial stiffness, and they could be used for BP measurement. However, the dicrotic wave would diminish in the PPG wave with age [47], and often be hidden in the main wave [48]. In our study, the dicrotic wave remained significant in the PPW signals from all the participants (age range: 49.74 ± 8.12 years), and the features extracted from the dicrotic wave of PPW, such as TmAE and TmBE, demonstrated favorable performances for BP estimation.

Feature extraction from the original pulse wave and the derivative of pulse wave for BP estimation has been commonly used in the literature [9,14,49]. The pulse characteristic value K was revealed to quantify vascular elasticity [50] and PIR extracted from PPG indicators can reflect the arterial diameter change [8]. Inspired by the literature, we calculated 21 features from PPW signals for BP measurement. Note that the goal of this study is to develop the BP estimation model from PPW as measured by piezoelectric based sensors. For evaluating the validity of the constructed PPW-based model, the BP estimation performance of the constructed PPW-based model was compared with that of the widely studied PTT-based model. Our results showed that the single PPW-indicator-based BP estimation model (model 1–21, as listed in Table 3) was less effective in predicting BP, since BP is the joint result of multiple factors. However, the MPF model, which was based on the fusion of twenty-one PPW-based models that exhibited superior performance in estimating BP. Compared with previous proposed PTT-based models [40,41,42,43], the MPF model shows better performance in terms of MD ± SD estimation errors under the same experimental scenario. The MPF model can reflect the comprehensive results of the PPW-based models; our results indicated that the proposed MPF model attained the most efficient BP estimation among the PPW-based models, and achieved a decrease in estimation errors of 1.33 ± 0.37 mmHg in SBP, and 1.14 ± 0.20 mmHg in DBP, in comparison with the PTT-based model. This result indicated that BP can be estimated accurately using PPW signals, which is meaningful for cuffless BP estimation in applications such as wearable devices. In addition, we evaluated the accuracy of the proposed MPF model for BP estimation at different calibration intervals to analyze the robustness of the proposed model. Experimental results indicated that the accuracy of BP estimation decreased as the calibration interval increased; however, with longer time intervals, the estimation error became relatively stable, which is similar to the findings of previous reports [10,37]. These results suggest a daily fluctuation existing in the proposed PPW-based method but not associated with a long-term degradation. Such daily degradation might result from uncontrolled experimental conditions such as measurement bias of the devices.

PPW is collected by using a piezoelectric sensor pressed on a single measuring point of the radial artery in our study, and thus may have suffered from high derivation. Flexible pressure sensor arrays, which can be collected in a multipoint way by affixing a group of flexible sensors to the skin, have great potential to substitute existing PPW sensors [21,22]. The use of a flexible pressure sensor array to measure PPW would make PPW acquisition more accurate and stable for noninvasive and cuffless BP estimation in wearable devices.

### 5.2. Limitations

Although this study validated the feasibility of PPW in cuffless BP estimation, some limitations require further investigation to ensure that the method is fully applicable. First, to better reconstruct the relationship between PPW and BP, the experiment was designed in a controlled scenario, with subjects required to lie down quietly for 16 measurements in the calibration process to avoid motion artifacts. Simplified experimental procedures should be considered in reality; for example, to reduce the number of measurements for training, and to allow the subjects to be in a sitting position. Second, the PPW signals were obtained using a piezoelectric pressure sensor placed on a single measuring point of the radial artery, which was complicated to operate. However, we believe that it can be operated in a wearable and stable way with the development of a flexible pressure sensor array. In addition, due to the limited number of samples in the training dataset, linear regression was used in our study. Nonlinear regression algorithms might improve the performance if enough samples are available.

## 6. Conclusions and Future Work

It this study, we evaluated the potential of utilizing a piezoelectric-induced sensor to measure PPW for BP estimation method in a cuff-less, noninvasive manner on a signal site of one hand. Based on pulse wave analysis, 21 informative features calculated using PPW that potentially influence BP were analyzed and used to develop a PPW-based BP estimation model, which was validated experimentally. A maximum 60-day follow-up study was conducted on 65 middle-aged and elderly participants, and the experimental results revealed that the PPW-based model achieved an estimation error of 0.7 ± 7.78 mmHg for SBP estimation, and 0.83 ± 5.45 mmHg for DBP estimation in terms of MD ± SD. Compared with the PTT-based model, the proposed model achieved higher accuracy, with a decrease of 1.33 ± 0.37 mmHg in SBP and 1.14 ± 0.20 mmHg in DBP in terms of estimation errors. This study thus demonstrated that PPW obtained from piezoelectric sensor could provide an effective method for cuffless BP estimation and could potentially be widely used in wearable devices for BP management.

In our future work, we will focus on using a simpler and more effective sensor to collect PPW signals and to validate the accuracy of the PPW-based method for BP measurement further, with more participants to make the PPW-based model more reliable. Furthermore, some machine-learning techniques such as artificial neural networks [12], support vector machines [51], and random forests [52] have been reported to have the potential for BP measurement. Therefore, we will attempt to validate the application of machine-learning techniques in selecting optimal sets of PPW features, and then establish a PPW-based BP estimation model with higher accuracy.

## Figures and Tables

**Figure 1 sensors-18-04227-f001:**
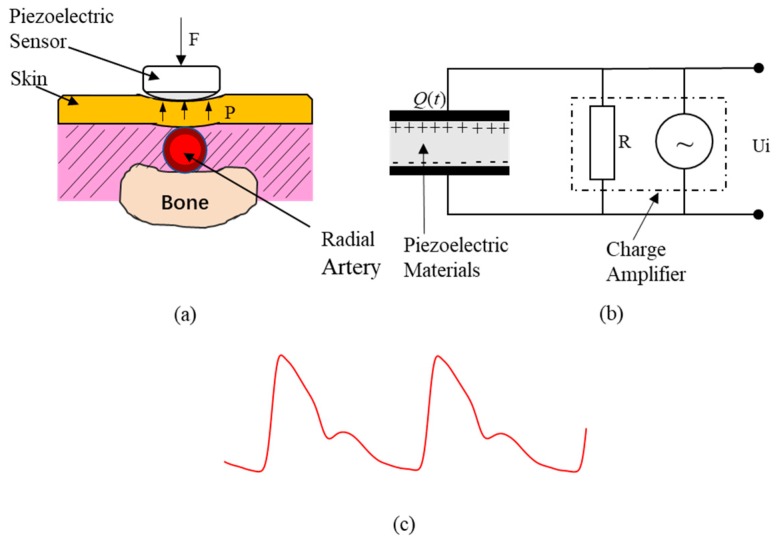
PPW measurement based on piezoelectric sensor. (**a**) Pressure pulse wave (PPW) measurement. (**b**) Pressure conversion of an electric signal model of a piezoelectric sensor. (**c**) PPW signals in radial artery.

**Figure 2 sensors-18-04227-f002:**
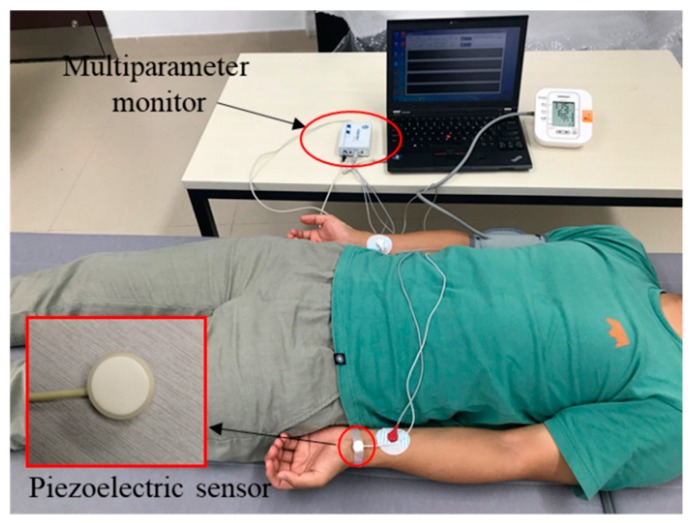
Experimental scenario.

**Figure 3 sensors-18-04227-f003:**
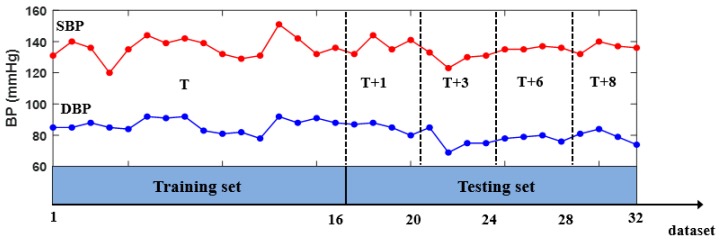
Experimental procedures at day T, T + 1, T + 3, T + 6, and T+8.

**Figure 4 sensors-18-04227-f004:**
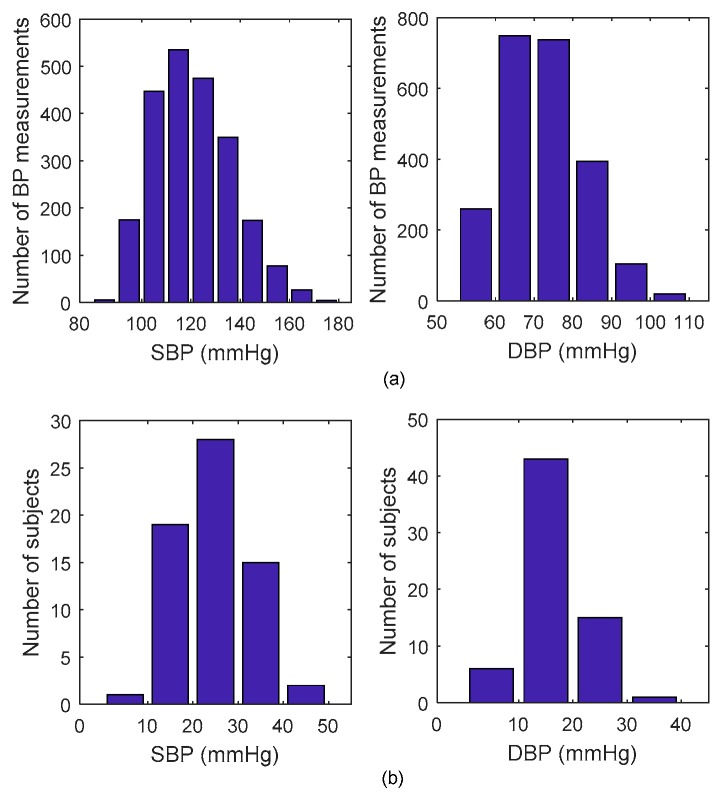
Statistical information of the systolic blood pressure (SBP) and diastolic blood pressure (DBP). (**a**) Statistical distribution. (**b**) Individual dynamic range distribution.

**Figure 5 sensors-18-04227-f005:**
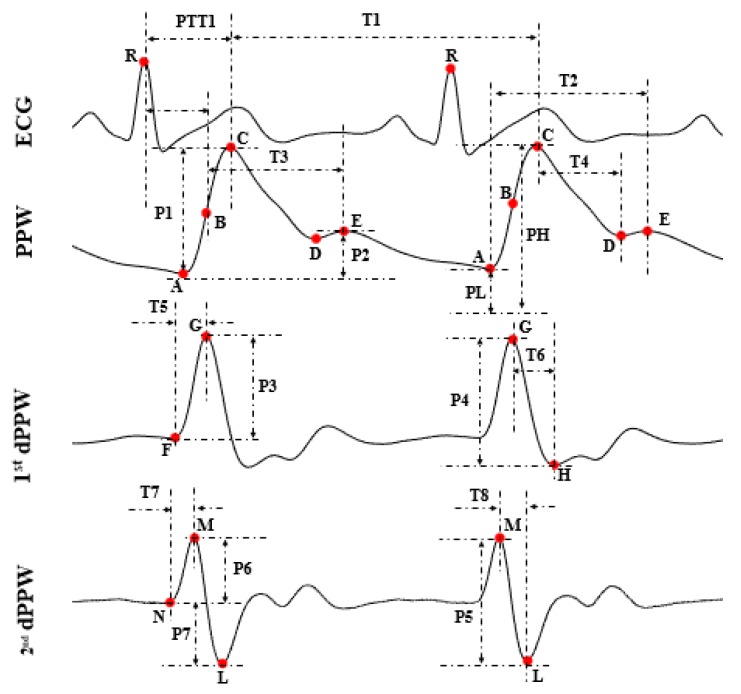
Extracted features.

**Figure 6 sensors-18-04227-f006:**
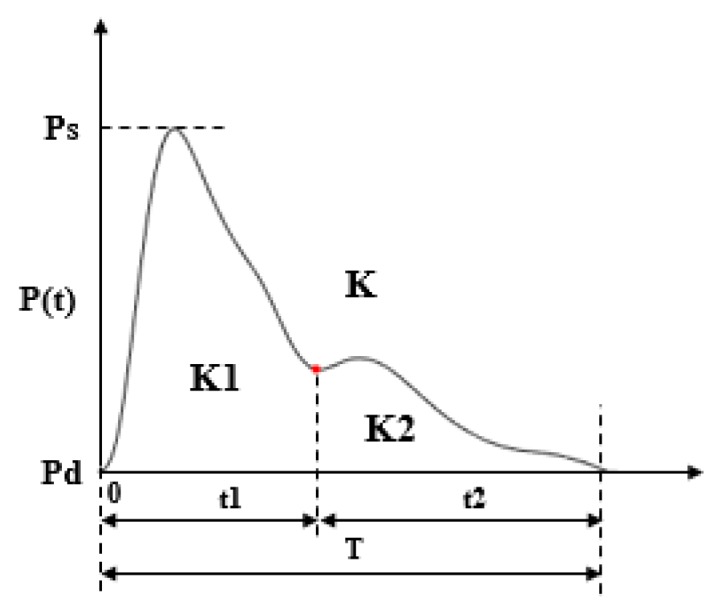
Definition of K-value in PPW.

**Figure 7 sensors-18-04227-f007:**
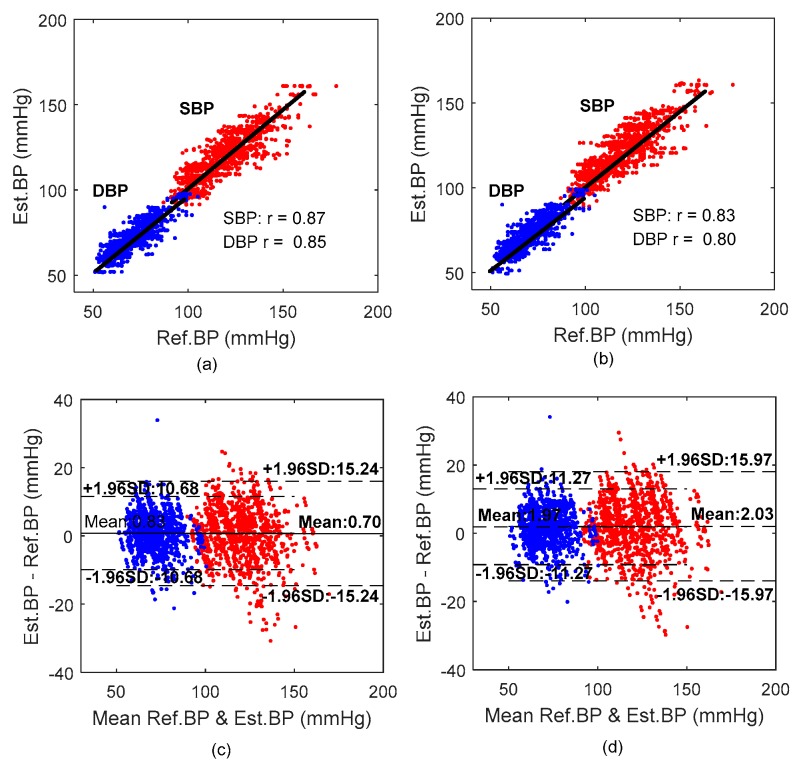
Correlation and Bland–Altman plots of SBP and DBP with the reference cuff-based BP. (**a**,**b**) Correlation between the reference and estimated BP. (**c**,**d**) Bland–Altman plot BP estimation. (**a**,**c**) From MPF-based model. (**b**,**d**) From PTT-based model.

**Figure 8 sensors-18-04227-f008:**
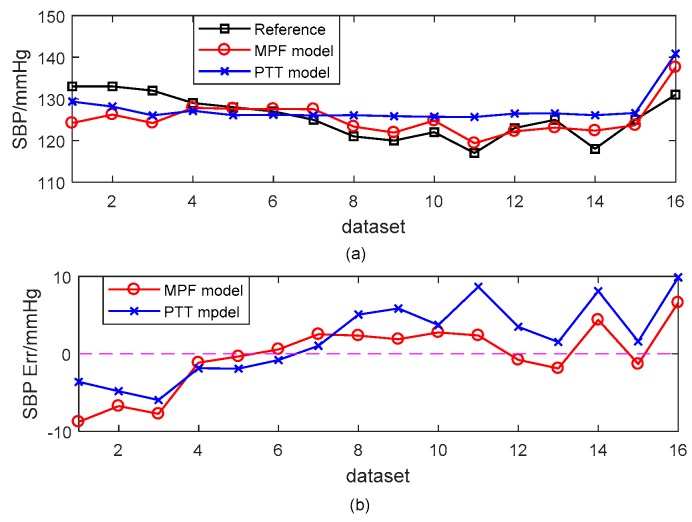
An example of the estimated SBP with the MPF model and the PTT-based model with the cuff-based BP as a reference. (**a**) Dataset SBP. (**b**) SBP errors.

**Figure 9 sensors-18-04227-f009:**
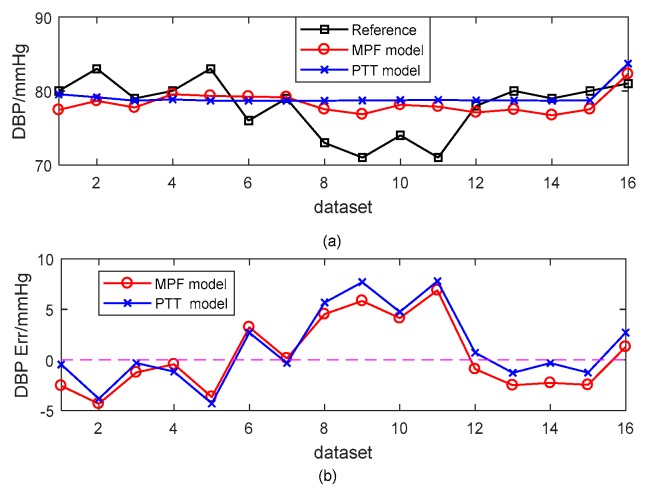
An example of the estimated DBP with the MPF model and the PTT-based model with the cuff-based BP as a reference. (**a**) Dataset DBP. (**b**) DBP errors.

**Figure 10 sensors-18-04227-f010:**
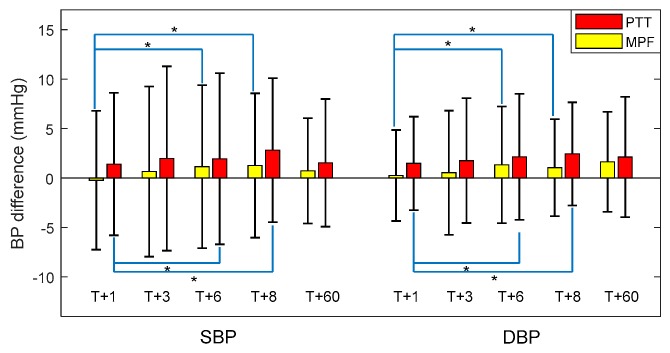
Estimation error for SBP and DBP at different time intervals (* indicates statistical significance at 0.05).

**Table 1 sensors-18-04227-t001:** Technique parameter summary of the HK-2000B pulse sensor.

Parameter	Definition
Pressure range	−50 to +300 mmHg
Pressure sensitivity	2000 μA/mmHg
Temperature coefficient	1 × 10^−4^ °C
Response time	<0.4 ms
Precision	0.5%

**Table 2 sensors-18-04227-t002:** Definitions of the selected features in PPW.

Features	Definitions	Equations
RtAmCE	Amplitude ratio of point C and point E	P2/P1
TmAE	Time span between point A and point E	T2
TmBE	Time span between point B and point E	T + 3
TmCD	Time span between point C and point D	T4
RtTP	Time ratio of T4 to peak interval	T4/T + 1
K	PPW characteristic value	Formula (1)
K1	Systolic characteristic value	Formula (2)
K2	Diastolic characteristic value	Formula (3)
AS	Ascending slope of PPWr	$\sum_{i=A}^{C}\left(P_{i}-P_{A}\right)$
1st dPPW_PAm	Peak Amplitude of 1st dPPWr	P3
1st dPPW_TW	Time width of 1st dPPWr	T + 6
2nd dPPW_TAm	Total Amplitude of 2nd dPPWr	P5
2nd dPPW_PAm	Peak Amplitude of 2nd dPPWr	P6
2nd dPPW_FAm	Foot Amplitude of 2nd dPPWr	P7
1st dPPW_AS	Ascending slope of 1st dPPWr	P3/T5
1st dPPW_DS	Descending slope of 1st dPPWr	P4/T + 6
1st dPPW_AA	Ascending area of 1st dPPWr	$\sum_{i=F}^{C}\left(P_{i}^{\prime }-P_{F}^{\prime }\right)$
2nd dPPW_AS	Ascending slope of 2nd dPPWr	P6/T7
2nd dPPW_DS	Descending slope of 2nd dPPW	P5/P8
2nd dPPW_AA	Ascending ared of 2nd dPPW	$\sum_{i=N}^{M}i=(P_{i}^{\prime \prime }-P_{N}^{\prime \prime })$
PIR	Ratio of PPW peak amplitude to foot amplitude	PL/PH
PTT	Time span between the ECG R peak and 1st dPPW peak	PT + 1

**Table 3 sensors-18-04227-t003:** Estimated BP errors of the PPW and PTT-based models.

		Estimated Error (MD ± SD) (mmHg)				Estimated Error (MD ± SD) (mmHg)	
Models	Variables	SBP	DBP	Models	Variables	SBP	DBP
1	RtAmCE	2.15 ± 8.45	1.76 ± 5.58	14	2nd dPPW_FAm	2.27 ± 8.07	2.03 ± 5.76
2	TmAE	2.34 ± 7.98	2.01 ± 5.65	15	1st dPPW_AS	2.19 ± 8.01	1.97 ± 5.59
3	TmBE	2.24 ± 8.10	1.88 ± 5.75	16	1st dPPW_DS	2.48 ± 8.08	2.16 ± 5.68
4	TmCD	2.39 ± 8.25	1.99 ± 5.89	17	1st dPPW_AA	2.06 ± 8.02	1.84 ± 5.54
5	RtTP	2.11 ± 8.00	1.88 ± 5.66	18	2nd dPPW_AS	2.11 ± 8.06	1.88 ± 5.75
6	K	2.27 ± 8.27	2.03 ± 5.82	19	2nd dPPW_DS	2.32 ± 8.10	2.17 ± 5.62
7	K1	2.22 ± 8.36	1.87 ± 5.75	20	2nd dPPW_AA	2.32 ± 8.03	2.03 ± 5.65
8	K2	2.40 ± 8.57	2.06 ± 5.95	21	PIR	2.21 ± 8.07	1.95 ± 5.74
9	AS	2.17 ± 8.14	1.92 ± 5.80	22	MPF	0.70 ± 7.78 ^a^	0.83 ± 5.45 ^b^
10	1st dPPW_PAm	2.15 ± 8.08	1.95 ± 5.76	23	PTT	2.17 ± 8.26	2.07 ± 5.71
11	1st dPPW_TW	2.15 ± 8.04	1.93 ± 5.71	24	1/PTT	2.22 ± 8.50	2.08 ± 5.70
12	2nd dPPW_TAm	2.31 ± 8.41	1.99 ± 5.86	25	ln(1/PTT), 1/PTT^2^	2.03 ± 8.15 ^a^	1.97 ± 5.75 ^b^
13	2nd dPPW_PAm	2.22 ± 8.11	1.99 ± 5.74				
Decrease estimation error						1.33 ± 0.37	1.14 ± 0.20

^a^ and ^b^ indicates statistical significance at the level 0.0001.

**Table 4 sensors-18-04227-t004:** Accuracy evaluation based on the British Hypertension Society (BHS) standard.

		CP at ± 5 mmHg	CP at ± 10 mmHg	CP at ± 15 mmHg	Grade
Proposed model (model 22)	SBP	50.95%	81.18%	94.77%	B
	DBP	64.45%	93.44%	98.76%	A
PTT-based model (model 25)	SBP	47.53%	77.28%	93.16%	C
	DBP	58.84%	89.62%	97.95%	B

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
