# Peer review of "Cuffless Blood Pressure Estimation Using Pressure Pulse Wave Signals"

_sensors, 2018, doi:10.3390/s18124227_

Round 1
Reviewer 1 Report
Nice piece of work, try not to include so many abbreviation in the abstract even though if you explain them. You definitely need to revise the reference appearance row in Introduction, because they are mixed up, like in lines 63-64 of your paper, reference 9 for example.
Author Response
Point 1: Nice piece of work, try not to include so many abbreviations in the abstract even though if you explain them. You definitely need to revise the reference appearance row in Introduction, because they are mixed up, like in lines 63-64 of your paper, reference 9 for example.
Response 1: Thanks a lot for your approval. We have reduced the number of abbreviations in the abstract section according to your kind suggestion. In addition, we have formatted all the reference according to the requirement of the journal carefully.

Reviewer 2 Report
In the manuscript, the authors introduced a new method based on PPW to calculate blood pressure for cuffless blood pressure measurement. Currently, cuffless, continuous, noninvasive method for measuring blood pressure is very important issue in cardiology. It is interesting that the author developed a method to calculate the blood pressure from PPW. I believe this research is an important paper that initiate the utilization of PPW for BP measurement. However, following issues are needed to be addressed before it can be published.
Major issues:
1. In the introduction, the authors commented that PTT–based model as alternative method of measuring BP. However, PAT (pulse arrival time) is another popular and convenient method to measure the time delay and to estimate blood pressure. For robust evaluation of the method based on PPW with conventional method, I suggest the authors to investigate the results from PAT-based method.
2. In page 8 line 225, the authors chose three PTT-based model to evaluate the performance of PWV based BP measurement. In table 2, the results from the model are displayed. It is important to check that the results are comparable to the previous studies that the authors cited from ref 39-42 for fair evaluation.
3. In Page 2 line 77, the authors said “However, few studies exist regarding the utility of PPW for cuffless BP estimation.” Please include the references of the studies and discuss the difference of their research from the previous study that utilized PPW for BP estimation.
Minor issues:
4. In Page 4 line 149, the authors used commercial piezoelectric sensor (HK-2000B, Hefei Huake 149 Electronic Technology Research Institute, China). The detail information of the sensor is needed for the reader to understand the reason why the sensor is used for this research such as gauge factor and response time.
5. In page 12 line 350 and figure 11 presented a flexible pressure sensor array. But, the authors did not use it for this study and there is no detail of the sensor. So, including the figure in the manuscript is not appropriate. I recommend to remove the figure in the manuscript.
Author Response
We are very appreciated for your approval on the improvement on previous submission and constructive comments, which are all valuable and very helpful for revising and improving our paper. We have studied all the comments carefully and have made revision which we hope meet with approval. For your convenience, our revised portion are marked in red font in the new submission, hope you will be satisfied with this revision. Looking forward to hearing from you about the further decision on our submission.
Point 1: In the introduction, the authors commented that PTT–based model as alternative method of measuring BP. However, PAT (pulse arrival time) is another popular and convenient method to measure the time delay and to estimate blood pressure. For robust evaluation of the method based on PPW with conventional method, I suggest the authors to investigate the results from PAT-based method.
Response 1: Thanks very much for your careful review. We are very sorry for making you confused about the PTT and PAT. We admit that PAT is a also popular and convenient method to measure the time delay between R-ECG and pulse wave to estimate BP. In fact, “PTT” in our previous submission also means the time delay between R-ECG and pressure pulse wave. There are distinct differences between research groups regarding the definition of PTT, and PTT has been usually mixed up with PAT. Why we choose “PTT” instead of “PAT” to describe the time delay? Actually, the usage of PTT can be dated back to 1964, when Weltman et al. [1] devised the PWV computer by “utilizing the ECG complex and a downstream pulse signal to define pulse transit time over a known arterial length”. To the best of our knowledge, this is the first time the word “pulse transit time” being used and defined. Except for early studies, current research about PTT from both medical and engineering perspectives [1-6] have accepted and calculated PTT as the time delay from R-ECG to the pulse (e.g., pressure pulse wave).
By 2015, our team had done literature survey with Thomson Reuters Web of Science, and found that the total number of references for “pulse transit time” was 270, with the sum of times cited was 2380, among which PTT was commonly used not only as the time difference from R wave of ECG to peripheral pulse but also as the time intervals between two peripheral pulses; while the total number of references for “pulse arrival time” was 49 with the sum of times cited was 396. As shown in Fig. 1 (The Fig.1 is in the attachment of response), the number of publications in the recent 20 years for “PTT” is increasing with years more than “PAT”. In addition, among 270 publications with title including “pulse transit time”, PTT that defined and calculated as the time interval between R wave of ECG and peripheral pulse is dominant. Therefore, the time delay between R wave of ECG and peripheral pulse was named “PTT” in our study. To reflect your concerns, we described the calculation method of “PTT” in the new submission, which is also quoted here for your information:
“PTT was calculated as the time intervals from the ECG R-peak to the peak of the 1st dPPW in the same cardiac cycle [8].”
[1] G. Weltman, G. Sullivan, and D. Bredon, "The continuous measurement of arterial pulse wave velocity," Medical Electronics and Biological Engineering, vol. 2, pp.145-154, 1964.
[2] Steptoe A, Smulyan H & Gribbin B 1976. Pulse-Wave Velocity and Blood-Pressure Change Calibration and Applications. Psychophysiology, 13, 488-493.
[3] Newlin D B 1981. Relationships of Pulse Transmission Times to Pre-Ejection Period and Blood-Pressure. Psychophysiology, 18, 316-321.
[4] Mccarthy B M, C J Vaughan, B O'flynn, A Mathewson & C O Mathuna 2013. An examination of calibration intervals required for accurately tracking blood pressure using pulse transit time algorithms. Journal of Human Hypertension, 27, 744-750.
[5] Liu Q, B P Yan, C M Yu, Y T Zhang & C C Y Poon 2014. Attenuation of Systolic Blood Pressure and Pulse Transit Time Hysteresis during Exercise and Recovery in Cardiovascular Patients. IEEE Transactions on Biomedical Engineering, 61, 346-352.
[6] Patzak A, Y Mendoza, H Gesche & M Konermann 2015. Continuous blood pressure measurement using the pulse transit time: Comparison to intra-arterial measurement. Blood Pressure, 24, 217-221.
Point 2: In page 8 line 225, the authors chose three PTT-based model to evaluate the performance of PWV based BP measurement. In table 2, the results from the model are displayed. It is important to check that the results are comparable to the previous studies that the authors cited from ref 39-42 for fair evaluation.
Response 2: We appreciate your comments. Please note that the Ref. 39-42 in the previous submission correspond to Ref. 40-43 in the new submission. Actually, the three PTT-based models were developed according to the approaches proposed in Ref. 40-43 on our dataset. In detail, the BP estimation model that used the relative change in PTT as independent variable is from reference 40; the model was developed based on 1/PTT is from reference 41; the model was developed by using both ln(1/PTT) and 1/PTT2 as independent variables is from Ref. 42-43, which are described in the last paragraph in section 3.4.1. Even though the experimental results based on PTT-based models in our study are different from that reported in Ref. 40-43, the experimental procedure and the number of participants is also different from previous studies. Our study is designed for middle-aged or elderly population and followed up for a maximum 60 days while previous studies aimed for cardiovascular patients [40] or chronic diseases with less verification cycle [42], and healthy young subjects [41,43]. Our results showed that the proposed MPF models achieved better accuracy than PTT-based models proposed in Ref. 40-43 on the same dataset. To reflect your concerns, we included a detail description in the new submission. We quoted here for your information:
“Compared with previous proposed PTT-based models [40-43], the MPF model shows better performance in terms of MD ± SD estimation errors under the same experimental scenario.”
Point 3: In Page 2 line 77, the authors said “However, few studies exist regarding the utility of PPW for cuffless BP estimation.” Please include the references of the studies and discuss the difference of their research from the previous study that utilized PPW for BP estimation.
Response 3: Thanks very much for your careful review. The PPW has been used for diseases diagnosis and arterial stiffness evaluation [1-3]. However, to best of our knowledge, this is the first time to utilize the PPW for estimating BP based on PPW analysis (calculating features that potentially influenced BP from PPW). Even though the arterial tonometry that has been used for continuous BP estimation also can reflect the changes of pressure waveform, the principles between PPW and tonometry for estimating BP are different, and those differences have been discussed in the previous permission, which is also quoted here for your information:
“It is worthwhile mentioning that arterial tonometry can also be used to measure the BP waveform within the blood vessel, which need a controlled force to maintain the superficial artery in applanated state over time [6, 29]. However, applanation have proven to be difficult so that the measured waveform should be frequently calibrated in practice [30]. Different from arterial tonometry, piezoelectric sensors do not need to flatten the artery, therefore the contact pressure is not quite restricted for PPW collection.”
[1] Joshi, A.; Kulkarni, A.; Chandran, S.; Jayaraman, V. K.; Kulkarni, B. D. In Nadi Tarangini: a pulse based diagnostic system, Engineering in Medicine and Biology Society, 2007. Embs 2007. International Conference of the IEEE, 2007; 2007; p 2207.
[2] Chu-Chang, T.; Shing-Hong, L.; Jan-Yow, C.; Jian-Jung, C.; Wen-Miin, L., A novel noninvasive measurement technique for analyzing the pressure pulse waveform of the radial artery. IEEE transactions on bio-medical engineering 2008, 55, (1), 288-97.
[3] Wang, D.; Zhang, D.; Lu, G., A robust signal preprocessing framework for wrist pulse analysis. Biomedical Signal Processing & Control 2016, 23, 62-75.
Point 4: In Page 4 line 149, the authors used commercial piezoelectric sensor (HK-2000B, Hefei Huake 149 Electronic Technology Research Institute, China). The detail information of the sensor is needed for the reader to understand the reason why the sensor is used for this research such as gauge factor and response time.
Response 4: Thank you for pointing out this issue. To response your concerns, we have added a detail information of the sensor in new submission, which is also quoted here for your information:
“The HK-2000B is a medical pulse sensor which integrated the pressure-sensitive devices (PVDF piezoelectric membrane), sensitivity temperature compensation components, temperature sensing element and signal adjustment circuits, and it has been recognized by some research institutions [36]. The major technique parameter of HK-2000B pulse sensor was summarized in table 1.”
Point 5: In page 12 line 350 and figure 11 presented a flexible pressure sensor array. But the authors did not use it for this study and there is no detail of the sensor. So, including the figure in the manuscript is not appropriate. I recommend to remove the figure in the manuscript.
Response 5: Thank you for your kind suggestions. We have removed the figure in new submission.

Round 2
Reviewer 2 Report
I am happy to the responses from the authors.